# The widespread nature of Pack-TYPE transposons reveals their importance for plant genome evolution

Jack S. Gisby[1¤]*, Marco Catoni[1,2]*

1 School of Biosciences, University of Birmingham, Birmingham, United Kingdom, 2 Institute for Sustainable Plant Protection, National Research Council of Italy, Torino, Italy

¤ Current address: Department of Immunology and Inflammation, Imperial College London, London, United Kingdom.
* j.gisby20@imperial.ac.uk (JSG); m.catoni@bham.ac.uk (MC)

## Abstract

Pack-TYPE transposable elements (TEs) are a group of non-autonomous DNA transposons found in plants. These elements can efficiently capture and shuffle coding DNA across the host genome, accelerating the evolution of genes. Despite their relevance for plant genome plasticity, the detection and study of Pack-TYPE TEs are challenging due to the high similarity these elements have with genes. Here, we produced an automated annotation pipeline designed to study Pack-TYPE elements and used it to successfully annotate and analyse more than 10,000 new Pack-TYPE TEs in the rice and maize genomes. Our analysis indicates that Pack-TYPE TEs are an abundant and heterogeneous group of elements. We found that these elements are associated with all main superfamilies of Class II DNA transposons in plants and likely share a similar mechanism to capture new chromosomal DNA sequences. Furthermore, we report examples of the direct contribution of these TEs to coding genes, suggesting a generalised and extensive role of Pack-TYPE TEs in plant genome evolution.

## Author summary

Transposable Elements (TEs) are genetic DNA sequences able to move across the genome, and their transposition activity is associated with genome plasticity and gene evolution. However, most of these elements exhibit "selfish" behaviour, meaning that they mainly transpose their own DNA sequence and only exceptionally might rearrange the DNA of coding genes. Pack-TYPE TEs, found in plants, represent an important exception, and they can efficiently capture and shuffle DNA sequences captured from the genome, accelerating the evolution of genes. We provide here the first automatic pipeline designed explicitly for the annotation of Pack-TYPE TEs. We used our approach to systematically investigate Pack-TYPE TEs in the rice and maize reference genomes, and annotated thousands of new elements in these species. We demonstrate that Pack-TYPE elements are

**Data Availability Statement:** The entire set of annotated TEs are available from S1 Table (Tab A, B, D and H). These annotations were generated using packFinder v1.2.0, available as part of the Bioconductor project (https://doi.org/doi:10.18129/

B9.bioc.packFinder). The code for the package may be found on GitHub (https://github.com/jackgisby/packFinder).

**Funding:** MC work is partially funded by the Royal Society Research Grant [RGS\R1\201297] (https://royalsociety.org/). The funders had no role in study design, data collection and analysis, decision to publish, or preparation of the manuscript.

abundant in plants and we report several examples of coding genes originated as a consequence of the mobilization of these elements.

## Introduction

Class II DNA transposable elements (TEs) are genomic loci able to move their sequence and relocate it to a new chromosomal location. This mobilisation process is mediated by a transposase typically encoded in the TE sequence, able to specifically recognise Terminal Inverted Repeat DNA sequences (TIRs) located at each end of the TE [1–3]. TEs that lack a functional transposase gene are defined as non-autonomous, and they can transpose if a transposase is provided *in trans* by a related autonomous element, as long as they maintain functional TIRs [4]. Due to reduced constraints on their DNA sequence, non-autonomous TEs can often become more abundant than their corresponding autonomous counterpart [5,6].

Pack-TYPE elements are a kind of Class II non-autonomous TE discovered in plants that contain DNA fragments captured from coding genes between their TIR sequences [7,8]. During mobilization, these elements can duplicate and transfer the captured coding DNA in different chromosomal locations, a process called transduplication [2]. In addition, if Pack-TYPE TEs insert into or in the proximity of genes, they have the potential to generate new transcriptional variants by merging their coding DNA with neighbouring exons, contributing to increased gene diversity [7,9].

In rice (*Oryza sativa*), more than 3000 Pack-TYPE TEs were annotated and classified as Pack-MULEs due to their TIR homology to autonomous elements belonging to the *Mutator*-like (*MULE*) superfamily [8]. However, the presence of coding gene fragments and the high variability in the sequence of these elements represent a challenge for automatic TE annotation tools. Although Pack-MULEs have since been found in other plants [10,11], manually curated approaches were required to identify their sequences in the genomes.

Recently, in *Arabidopsis thaliana*, a new family of Pack-TYPE elements has been discovered with TIRs similar to autonomous elements of the *CACTA* superfamily, which were therefore defined Pack-CACTA [7]. In contrast to Pack-MULEs, Pack-CACTAs mobilise in Arabidopsis epigenetic recombinant inbred lines, and their study clarified their transposition process and the mechanism used to acquire new coding DNA [7]. Although it was unclear how common Pack-CACTAs were in plant genomes, their discovery in Arabidopsis demonstrated that *MULE* is not the only superfamily with Pack-TYPE TEs; this raised the question of whether other superfamilies of TEs with TIRs could potentially support the transposition of compatible Pack-TYPE elements [12].

Here, we use common features of Pack-TYPE transposons to automatically annotate these elements in plant genomes, providing a specific and reliable tool for their annotation and clustering. In addition, we show that Pack-TYPE TEs are not limited to the *MULE* and *CACTA* superfamilies, and we found examples of exon shuffling events mediated by Pack-TYPE TEs belonging to all main superfamilies of plant DNA transposons with TIRs.

## Results

### Automatic annotation of Pack-TYPE TEs

In previous analyses, Pack-MULE and Pack-CACTA elements inserted respectively in the *Oryza sativa* and *Arabidopsis thaliana* reference genomes were identified using a combination of BLAST and manual annotation [7,8]. In order to optimise and automate the annotation of

Pack-TYPE transposons for metagenomics studies, we standardised the detection procedure in an algorithm implemented in the R package *packFinder* (**Fig 1A**) [13]. The algorithm takes as input TIR sequences and categorises the identified putative TEs according to the functional annotation previously applied to Pack-MULEs [10], which consists of three groups: i) autonomous TEs (with a valid blast hit to a transposase); ii) Pack-TYPE TEs (with a valid BLAST hit to coding genes); iii) non-Pack-TYPE TEs (remaining elements without significant blast hits to DNA transposases or coding genes).

Using as input the TIRs of the *Arabidopsis thaliana* autonomous *CACTA* elements ("**CAC-TACAA**") [14], we annotated 54 CACTA-like elements grouped in 12 clusters in the *Arabidopsis thaliana* reference genome (**Tab A in S1 Table**). This list includes the three previously identified Pack-CACTA families with intact TSD sequences [7] annotated in our analysis as Pack-CACTA in clusters #31 and #35 and as non-Pack TE in cluster #40. In addition, we annotated two more clusters (#28 and #39) of non-Pack TEs not previously annotated. The algorithm also found 7 clusters (#2, #6, #9, #10, #14, #22 and #23) of potentially autonomous *CACTA* TE families. All autonomous TEs identified were already annotated in the TAIR10 database as members of the *CACTA* superfamily (*ATENSPM* in Arabidopsis) (**Fig 1B**).

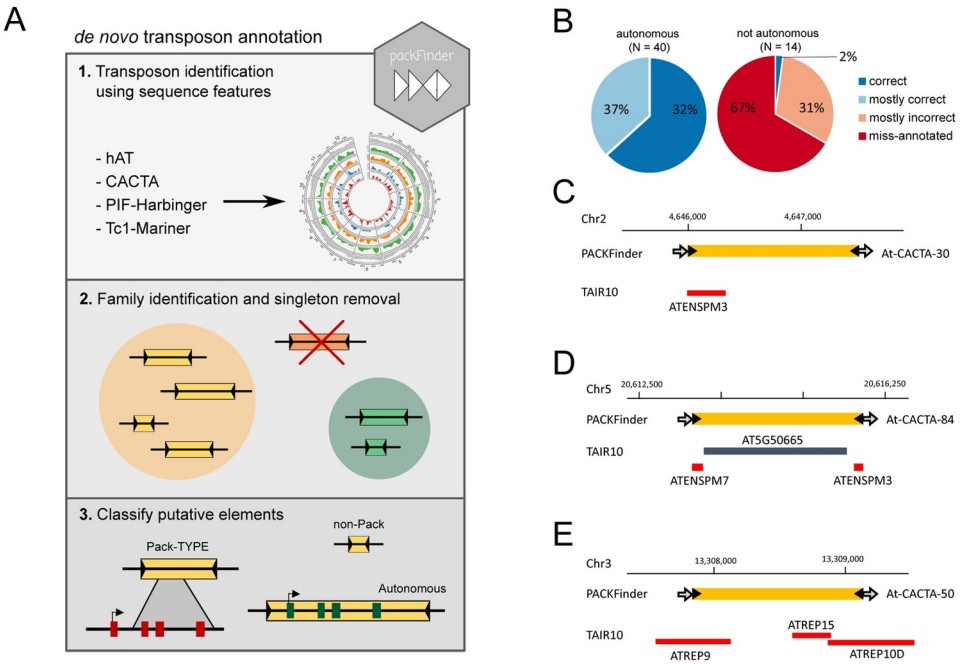

**Fig 1. The *packFinder* automatic annotation pipeline identifies Pack-CACTA TEs in Arabidopsis. A** The overall *packFinder* annotation stages. *packFinder* uses sequence features (TIRs and TSDs) to identify transposable elements; clustering is employed to group elements into sub-families and remove singleton elements that could represent false-positive annotations. Finally, BLAST is used to classify putative elements based on sequence identity to known transposases or other genes (see **Methods**). A more extensive vignette, including code, is available from Bioconductor (https://bioconductor.org/packages/release/bioc/html/packFinder.html). **B** Pie charts displaying the quality of TAIR10 annotation of autonomous and not autonomous (Pack-TYPE and not Pack) TEs identified by the *packFinder* automatic pipeline. The TAIR annotations have been classified as: i) correct, if the TE is annotated identically by TAIR10 and *packFinder*; ii) mostly correct, if in TAIR10 the TE is not uniquely identified, but > 50% of its sequence is annotated as belonging to *CACTA* (*ATENSPM*) superfamily; iii) mostly incorrect, if in TAIR10 less than 50% of TE sequence is annotated as belonging to *CACTA* (*ATENSPM*) superfamily; or iv) misannotated, if in TAIR10 the TE is not annotated or incorrectly annotated as a coding gene or as belonging to an unrelated repeat. **C, D** Examples of incorrect TAIR10 annotation or **E** misannotation of Pack and non-Pack TEs. TAIR10 annotation as genes (blue box) and TEs (red boxes) is displayed, and the corresponding gene name or TE family is reported. *packFinder* annotations (yellow boxes) are displayed with TIRs (black arrow) and TSDs (grey arrows).

However, some Pack-CACTA and non-Pack-TYPE annotations in TAIR10 were inaccurate (i.e. with only a portion of their sequence recognised as *CACTA* element) (**Fig 1C**), while others were misannotated as a protein coding gene (**Fig 1D**) or as TE repeats not belonging to the *CACTA* family (but mostly as *ATREP* and *HELITRON* families) (**Fig 1E**). This indicates that, compared to autonomous transposons, TAIR10 annotation could be less accurate for Pack-CACTAs or other *CACTA*-derived non-autonomous elements. Instead, our automatic algorithm, using only TIR sequences and the Arabidopsis genome as input, correctly annotated intact Pack-CACTA TEs previously manually identified in Arabidopsis, detecting also additional non-Pack-TYPE elements (**Tab A in S1 Table**).

## The *packFinder* pipeline can reliably annotate Pack-TYPE *Mutator* elements

Considering that the Arabidopsis genome is small and contains only a few Pack-CACTA TEs, we tested *packFinder* by re-annotating Pack-MULEs, which are an abundant superfamily of Pack-TYPE TEs well-characterized in the rice genome [8]. We input the 186 *MULE* terminal sequences used for the original Pack-MULE annotation to screen for TEs using *packFinder* [8]. We separately tested the effect of decreasing the allowable mismatches for TIR and TSD sequences on the number of false negatives. When TSD presence was ignored, *packFinder* could recover 95% of Pack-MULEs when ten base mismatches were allowed for TIR recognition (**S1A Fig**). Conversely, when testing only for the effect of allowable TSD mismatches on identification rates, two mismatches were enough to annotate >99% (2740) of the original Pack-MULEs (**S1B Fig**). These results indicate that an appropriate set of mismatches values can effectively control the level of false negatives.

Then, to evaluate the tool's overall performance, we calculated the false-positive rate of the automated algorithm as the number of allowable TIR mismatches varies. To do this, we repeated the *packFinder* search by inverting forward and reverse TIRs, so that the algorithm returned TIRs facing outwards rather than inwards. Considering that TIRs facing outwards are unlikely to be part of the same TE, their occurrence in the genome can be assumed to be equivalent to the general probability of false matches. We used these data to generate a receiver operating characteristic (ROC) curve (**S1C Fig,** see **Methods**) with an associated area under the curve (AUC) of 0.85, demonstrating good performance of the *packFinder* algorithm for identifying Pack-MULEs in rice. Allowing 7 TIR mismatches and 2 TSD mismatches resulted in our algorithm identifying a total of 8348 potential *MULE*-like elements (**Tab B in S1 Table**), including 2068 (75%) of the rice Pack-MULEs with a minimal (<1%) false-positive rate.

## Pack-TYPEs are prevalent across Class II TE superfamilies

Except for the *MULE* superfamily, other TIR TEs [6] have not been comprehensively analysed to identify Pack-TYPE elements. Therefore, to uncover the widespread nature of Pack-TYPE TEs, we investigated whether other DNA TE superfamilies (i.e. *hAT*, *Harbinger-PIF*, *Mariner*) could generate Pack-TYPE elements in rice (*Oryza sativa)* and maize (*Zea mays)*.

To *de novo* annotate *CACTA* elements, we used the conserved TIR sequences of rice *CACTA* and maize *En* TEs [14] in our automatic procedure to survey respectively the *Oryza sativa* and *Zea mays* reference genomes. To annotate *hAT* TEs, we used the core TIR sequences of the well characterised autonomous *Bg* and *Ac* [14,15] elements for maize and rice, respectively (**Tab C in S1 Table**). Similarly, for the *Harbinger-PIF* and *Mariner* superfamilies, we used the core TIR sequences from autonomous *PIF* and *Stow* (*Mariner*) elements [16,17] for both *Zea mays* and *Oryza sativa* (**Tab C in S1 Table**). We also applied our algorithm to annotate TEs in the *Arabidopsis thaliana* genome, using TIRs of elements previously described in

this species (**Tab C in S1 Table**). However, since we did not detect any new superfamilies of Pack-TE, we excluded this plant from subsequent analyses.

Contrary to Arabidopsis, we successfully annotated TEs for all tested superfamilies in maize and rice (**Tab D in S1 Table**), which appear to be distributed similarly in the genomes (**S2 Fig**). In maize, *hAT* and *Harbinger-PIF* elements were the most numerous (1,641 and 1,036, respectively), while we found a moderate number of *CACTA* elements (324 TEs) and relatively few *Mariner* (52 TEs). Conversely, in the rice genome, we found a high proportion of *Mariner* elements (642 TEs) and a moderate number of *Harbinger-PIF* elements (225 TEs), while *CACTA* and *hAT* TEs were of lower abundance (75 and 59 elements, respectively). Similarly to what we observed in Arabidopsis for *CACTA* TEs, the annotation of non-autonomous elements generally appeared to be less accurate than for autonomous TEs for all superfamilies considered, in both the rice and maize reference annotations (**S3 Fig**).

As expected, all TEs annotated in our analysis contain TIRs and a TSD, consistent with the features of the associated superfamily. Specifically, in both species, all the identified *hAT* TEs contained a TSD of 8 nt while the *CACTA* elements had TSDs of 5 nt, as described previously for these groups [14,15]. In addition, the TSDs of virtually all detected *Mariner* elements were "TA" in both *Oryza sativa* (99.1%) and *Zea mays* (98.1%), as previously described for this class of elements [16]. Similarly, the majority of identified *PIF* elements had "TTA" or "TAA" TSDs [17] in both *Oryza sativa* (52.4%) and *Zea mays* (81.2%). To further confirm the correct annotation of Pack-TYPE TEs, we aligned and clustered the first 80 bp of the forward TIR sequence and observed that, as expected, elements were grouped almost exclusively based on the superfamily to which they belong (**Fig 2A**). Interestingly, *PIF* and *CACTA* elements formed two independent clusters, suggesting the existence of sub-groups of TEs with different TIR sequences for these two superfamilies (**Fig 2A**). Collectively, these observations indicate that our pipeline successfully annotated TEs from different families of DNA transposons.

We then analysed the proportion of annotated Pack-TYPE elements in each superfamily and observed that their number varied depending on the superfamily and plant genome considered (**Fig 2B and Tab E in S1 Table**). Specifically, we observed that Pack-TYPE elements belonging to the *CACTA*, *hAT* and *PIF* superfamilies were more abundant in maize, while Pack-Mariner accumulated more in the rice genome. In addition, the proportion of Pack-TYPE elements we discovered was largely independent of the number of autonomous elements (or non-Pack TEs) found in the same superfamily (**Fig 2B and Tab E in S1 Table**). We also observed relatively high variability in the widths of the annotated Pack-TEs, with maize Pack-TYPE elements having a greater median width than rice elements for all annotated superfamilies (**Fig 2C**). In contrast, all TEs classified as non-Pack tended to be smaller than TEs classified as Pack-TYPE or autonomous in both plant genomes (**Fig 2C**). The cluster size distribution of the annotated TEs (**Fig 2D**) appears to be consistent across the *Oryza sativa* and *Zea mays* genomes, with the largest clusters (containing more than one hundred elements in maize) belonging to the *hAT* and *PIF* superfamilies (**Tab D in S1 Table**). By contrast, most *CACTA* and *Mariner* Pack-TYPE elements are low copy number in both species (**Fig 2D**).

## Pack-TYPE TEs acquire new DNA by mobilisation of neighbouring insertions

The frequent excision and insertion of DNA TEs in local chromosomal areas (also called "local hopping") have been proposed to facilitate the acquisition of new DNA and the evolution of Pack-*CACTA* TEs in Arabidopsis [7]. We searched for evidence of local hopping occurring during the mobilisation of Pack-TYPE TEs by investigating instances of local insertions in the set of annotated Pack-TYPE clusters.

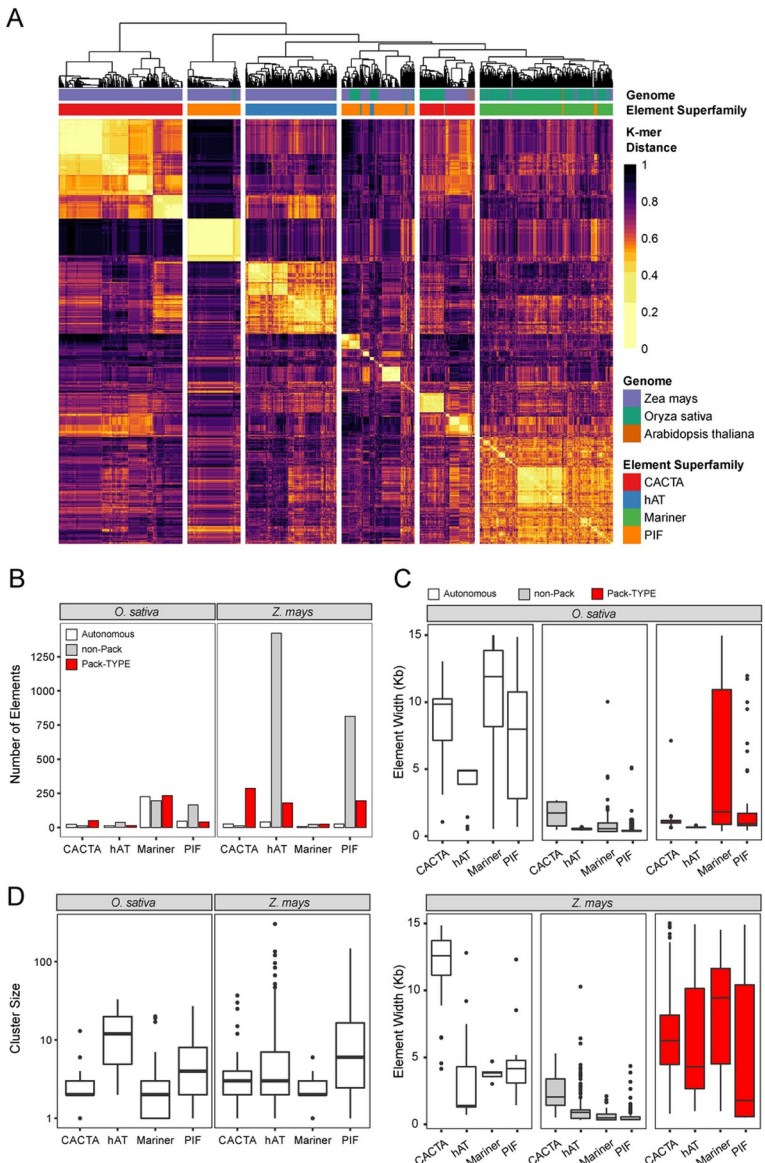

**Fig 2. The DNA sequence properties of Pack-TYPE TEs in *Oryza sativa* and *Zea mays*. A** Visualisation of the symmetric distance matrix for the forward TIRs of all annotated Pack-TYPE TEs. Distance was calculated using the alignment-free kmer algorithm, and elements are ordered by hierarchical clustering (see **Methods**). The dendrogram was cut such that six groups were identified, separated by vertical whitespace. **B** Bar plot displaying the total number of elements annotated by *packFinder* in each of the TIR superfamilies tested in both rice (*O. sativa*) and maize (*Z. mays*) genomes. Colours represent TE functional designation, assigned automatically using BLAST. **C** Box plots displaying the distribution of TE widths in each category, defined after applying the *packFinder* BLAST step in both the rice (*O. sativa*) and maize (*Z. mays*) genomes. **D** Boxplots for the distribution of annotated cluster sizes in both rice (*O. sativa*) and maize (*Z. mays*) genomes.

We defined local insertions as all pairs of annotated TEs belonging to the same cluster and within 100kb of distance from each other. In order to have a sufficient number of pairs to apply statistics, we focussed the analysis only on superfamilies with at least ten local insertions, which included the maize *hAT* (15 local insertions belonging to 8 clusters) and *PIF* (13 local insertions belonging to 5 clusters) superfamilies. For both groups, the aggregated proportion of observed local insertions of elements belonging to the same cluster was significantly greater

(for *hAT*-like p = 6.6x10$^{-12}$ and for PIF-like p = 2.9x10$^{-11}$) than the occurrence of local insertions of members of different clusters. We also observed that, among all local TE pairs considered, 80% (12 out of 15) for *hAT* and 92% (12 out of 13) for *PIF* were in the same orientation on the chromosome, suggesting that these TEs can influence the direction of insertion in their proximity for elements of the same family. Interestingly, local insertions with conserved orientation have been observed in real-time during the mobilisation of Pack-CACTA elements in Arabidopsis [7].

If two Pack-TYPE TEs with compatible TIRs are inserted within a short distance at the same locus, a transposase could recognise the two external TIRs as the start and end of a mobile element; and the two original TEs can then mobilise as a single element, including the DNA located initially between the two insertions [7] (**Fig 3A**). Therefore, Pack-TYPE TEs should contain newly acquired DNA located internally in phylogenetically related elements, while TIRs and terminal DNA sequences should be more conserved among elements of the same family. We evaluated this hypothesis by estimating the average uniqueness (calculated with the mappability, see **Methods**) of DNA sequences for each superfamily in the genome where most Pack-TYPE TEs were annotated, including Pack-Mariners in the rice genome and Pack-CACTAs, Pack-PIFs and Pack-hATs in the maize genome. We observed that all Pack-TYPEs tested contain less repeated DNA than all TEs annotated in rice and maize belonging to the corresponding superfamily (**S4 Fig**), and this effect tended to be more relevant in the internal part of the TEs (**S4 Fig**). This result is compatible with a model where more recently acquired coding DNA (assumed to be less repeated) is located more internally and further from TIR sequences.

We further investigated the structure of these Pack-TYPE TEs by alignment at the level of single clusters and found examples of DNA sequence acquisition events that are compatible with the fusion of neighbouring TE insertions. For example, in Zm-Pack-PIF cluster #18 in maize, we classified TEs into two groups based on their sequence similarity (**Fig 3B** and **Tab F in S1 Table**). Group A was composed of 52 non-Pack elements with similar lengths (average size of 402 bp), while Group B constituted 72 slightly longer Pack-TYPE TEs (with an average size of 535 bp). The main difference among elements of the two groups was an internal insertion of 163 bp found in all TEs in group B, with 97% similarity to the gene LOC109943976. In four elements of this group (subgroup B1), an additional insertion of 58 nucleotides was found, with 98% similarity to the gene LOC103626209 (**Fig 3C**).

In a second example, the Os-Pack-Mariner cluster #217 in rice, elements could be separated into four groups (**Fig 3D** and **Tab F in S1 Table**). Group A includes eight non-Pack TEs with an average length of 311 bp, while B contains three Pack-TYPE elements (that include a 280 bp DNA portion with 93% similarity to the rice gene Osg41365) and TIR sequences similar to group A (**Fig 3E**). Interestingly, three elements (Group AB) contain both the sequence of the A and B groups spaced by some additional DNA of unknown origin, while another two elements (group BB) have a similar structure but contain the sequence of group B repeated twice (**Fig 3E**). This suggests that two smaller TEs fused to generate elements of the AB and BB groups.

We observed similar examples of sequence acquisition events for Pack-TYPE TEs belonging to the *CACTA* and *hAT* superfamilies (**S5 Fig**). Collectively, these observations suggest that Pack-TYPE TEs of all DNA superfamilies can acquire DNA with a similar mechanism based on tandem insertions and re-mobilisation, as previously hypothesised for Pack-CACTA [7].

## Genes captured

We annotated Pack-TYPE elements using TIRs obtained from a set of known and well-characterized TEs (**Tab C in S1 Table**), which might not be representative of most Pack-TYPE TEs

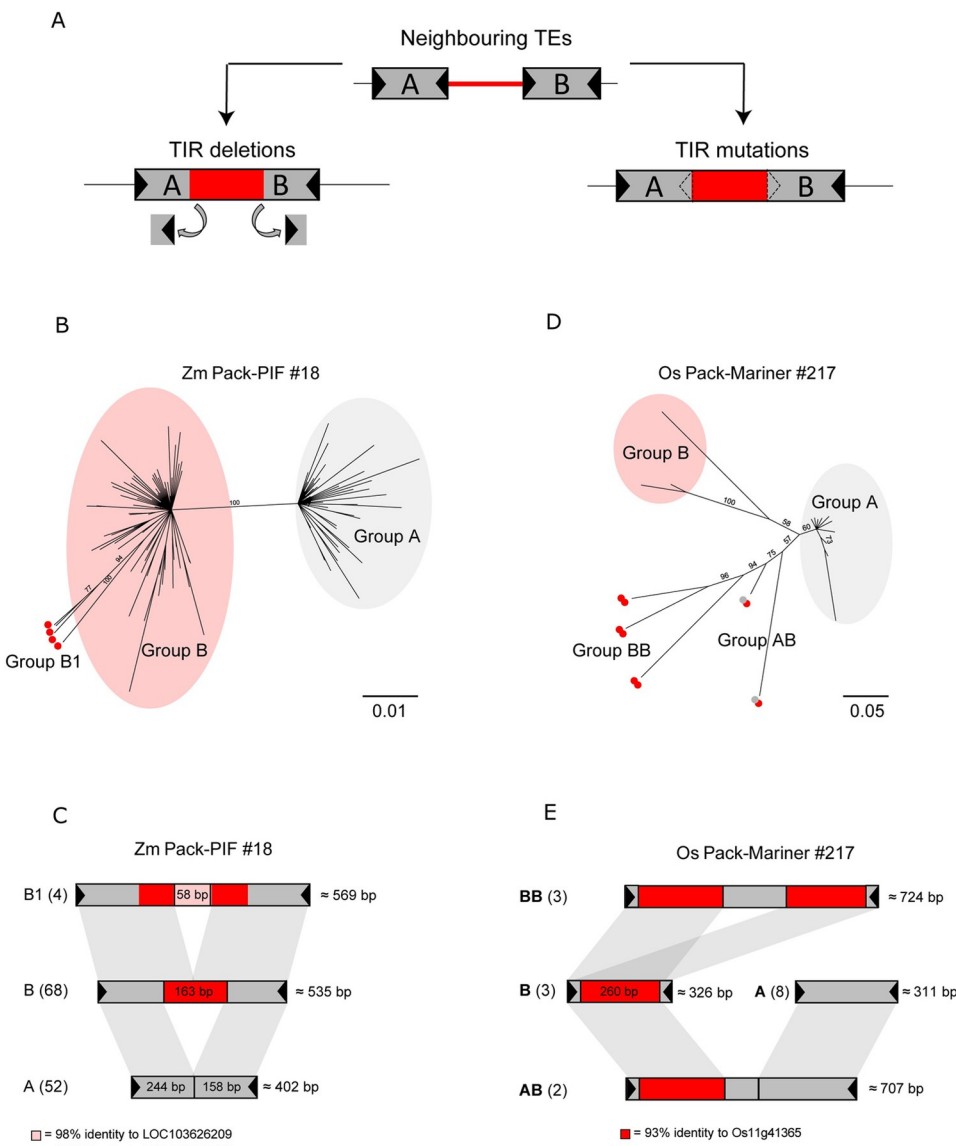

**Fig 3. Evidence of Pack-TYPE TE evolution in the maize and rice genomes. A** Scheme illustrating the model for the acquisition of chromosomal DNA by Pack-TYPE TEs (Catoni et al., 2019). A pair of neighbouring Pack-TYPE TE insertions (marked as A and B) and the DNA region between them (marked in red) can be recognised as single elements and mobilised by a transposase, either by deletion (left) or mutations (right) occurring at the internal TIRs. A black triangle represents TIRs. **B** Phylogenetic tree from the alignment of full-length sequences for elements in Zm-Pack-PIF cluster #18. Numbers at each node indicate bootstrap support values of 100 replications. Elements in groups and subgroups are shaded and marked with a red dot, respectively. **C** Structure of the Zm-Pack-PIF TE family #18. Each bar represents a group or subgroup as shown in **B** (number of elements per group indicated in parenthesis). Shaded grey marks regions with sequence homology. The average size of elements in each group is displayed on the right. In red is a 163 bp region with 97% identity to the maize gene LOC109943976, while pink marks a 58 bp region with 98% identity to the gene LOC103626209. The black triangles mark the TIRs (15 nt). **D** Phylogenetic tree from the alignment of full-length sequences of elements in Os-Pack-*Mariner* cluster #217. Numbers at each node indicate bootstrap support values of 100 replications. Elements in groups A and B are shaded in grey and red, respectively. TEs including two copies of elements in group B are marked with two red dots (Group BB), while elements including a copy of A and a copy of B are marked with a red and a grey dot (Group AB). **E** Structure of Os-Pack-Mariner TE family #217. Each bar represents a group shown in **D** (with the number of elements per group indicated in parenthesis). Shaded grey marks regions with sequence homology. The average size of elements in each group is displayed on the right. A 260 bp region with 93% identity to the rice gene Os11g41365 is shown in red. The black triangles mark the TIRs (15 nt).

in the rice and maize genomes. To comprehensively study Pack-TYPE elements and the coding DNA associated with them, we obtained a more exhaustive annotation of Pack-TYPE TEs using the most abundant core TIR sequences (10 bp) in the rice and maize genomes, screened with the homology-based TIR-Learner tool [18] (**Tab G in S1 Table**). Then, we used these TIRs as input to *packFinder* to re-annotate elements in the rice and maize genomes, detecting a more extensive set of putative transposons (3,183 and 40,692 in the rice and maize genomes, respectively) (**Tab H in S1 Table**). We found a limited overlap between the list of TEs identified by *packFinder* and the available TE annotations in the maize [19] and rice genomes [20] (**S6A Fig**). This suggests that, while *packFinder* did not reproduce the available comprehensive annotation of all TIR TEs in these genomes, it can identify several elements missed in previous annotations. Except for *CACTA* elements, *packFinder* tended to annotate TEs of larger sizes compared to the maize and rice annotations (**S6B and S6C Fig**).

Despite the greater number of TEs found with this approach, we observed similar properties to the previous set of annotated TEs (**Tab D in S1 Table**) in terms of the genome distribution of each superfamily (**S7 Fig**). In addition, when we aligned and clustered the first 80 bp of the forward TIRs, we still observed good separation by superfamily (**S8 Fig**). However, in maize, a large number of *hAT* and *PIF* elements could not be clearly discriminated (**S8A Fig**), suggesting either that the ten base pair TIR sequence was not sufficient to annotate these superfamilies or that TIRs used as input to *packFinder* for these elements were not correctly annotated in the reference maize genome. Nonetheless, the cluster sizes, the elements classified in each superfamily, and the proportion and the width of annotated Pack-TYPE TEs (**S9 Fig**) were similar to the results obtained for our original annotation (**Fig 2B, 2C and 2D**), confirming our previous observations and demonstrating consistency in the performance of *packFinder* using different sets of TIRs as input.

Having confirmed that the original and expanded sets of TEs had comparable properties, we took the union of the two sets to investigate captured genes' properties. Similar to the rice Pack-MULEs [8] and Arabidopsis Pack-CACTA [7], we also observed examples of Pack-TYPE elements containing genic material from multiple chromosomal loci (**Tab I in S1 Table**). We found that, in maize, Pack-TYPE TEs often contained coding DNA from more than one locus, with Pack-CACTA elements containing the greatest number of distinct loci (Mann-Whitney test, FDR <0.01). Conversely, in rice, a significant number of elements with multiple distinct loci (19%) were found only for Pack-Mariners (**Tab I in S1 Table**). We found that element width correlated (Pearson's) with the number of distinct loci captured (two-sided p<0.001 for each superfamily and genome), which indicates that larger TEs have the capacity to incorporate more chromosomal DNA.

We next sought to explore the expression and functional enrichment of the set of genes captured by Pack-TEs. We selected the best CDS match (by E-value) for each Pack-TYPE element to generate high-confidence lists of "captured" genes for each TE family and genome. We then identified significant (P-value <0.05) functional enrichments of GO terms using topGO [21] (**Tab J in S1 Table**). One of the most significantly enriched biological process (BP) terms, found in both maize and rice, was "post-embryonic development". We also found "flower development" and "anatomical structure morphogenesis" enriched for Pack-Mariner elements in rice. Other enriched terms involved response to endogenous stimulus and oxidative stress, protein localization and phosphorylation, and regulation of transcription. Interestingly, for maize, we also found enrichment of the term "DNA integration", suggesting that some Pack-TYPE TEs could still contain small fragments of transposase genes. Altogether, these results indicate a bias in the origin of coding DNA found in Pack-TYPE TEs.

To further investigate genes found in Pack-TYPE TEs, we obtained the developmental gene expression atlas publically available for both rice [22] and maize [23]. Then we used these

datasets to investigate the expression of genes "captured" in the most abundant Pack-TYPE superfamilies in each species (**S10A, S10B** and **S10C Fig**). We observed differences in the patterns of gene expression associated with different Pack-TYPE superfamilies. Specifically, in maize, we found that the genes captured by Pack-PIFs were expressed predominantly in leaves and green tissues (**S10A Fig**), while Pack-hAT were also included gene expressed in roots (**S10B Fig**) and among the genes captured by Pack-CACTAs were a set of genes expressed relatively highly in embryo tissues (**S10C Fig**). Most genes captured by elements in the rice genome were within Pack-Mariner elements. We observed that these genes were primarily expressed in seedlings but also in flowers and reproductive tissues, a result consistent with the enrichment of the "flower development" GO Term observed for this group (**Tab J in S1 Table**).

## Pack-TYPE TEs contribute to plant gene evolution

During mobilisation, Pack-MULE TEs shuffle exons across the genome, contributing to the generation of new transcript variants [8]. Genes that evolve in such a way can acquire large portions of coding DNA in a short evolutionary time, and homologs in other species can lack the DNA encoded by the sequence of the Pack-TYPE TEs [11]. To test this property in the newly identified Pack-TYPE superfamilies, we retrieved genes with coding sequences that overlapped the original set of Pack-TYPE TEs (**Tab K in S1 Table**). We then used the Plant Ensembl database [24] to identify homologous genes in other phylogenetically related genomes, and we compared syntenic genomic regions containing TE-derived DNA for one representative locus of each of the four Pack-TYPE superfamilies studied.

For example, the maize Zm-CACTA-181 TE is located in the 3-prime region of the gene Zm000021d052675, encoding the sequence of the last seven gene exons. However, in the syntenic genomic region of *Brachypodium distachyon*, the orthologous gene misses all exons encoded in the TE sequence (**Fig 4A**). Similarly, the Zm-Pack-PIF-686 element has inserted into the Zm00001d036443 locus, encoding the gene's second exon sequence. This insertion appears to be absent in the ortholog identified in *Setaria italica* (Si007332m.g, with ~80% sequence identity), which lacks the exon located in the TE (**Fig 4B**). A *Mariner* Pack-TYPE element (Os-Mariner-4) contributes the fourth exon of the gene LOC_Os01g04110 in the rice genome. Also, in this case, the TE is not present in the *B. distachyon* ortholog (BRADI_2g02180v3), where the exon encoded within the TE is missing (**Fig 4C**). Finally, the Zm-hAT-TL35758 element has inserted into the maize gene Zm00001d013112, extending the length of the sixth exon and providing a new splice donor site. In the syntenic DNA region of *Sorghum bicolor* (>84% sequence identity), the insertion of Zm-hAT-TL35758 is absent, and the orthologous gene's exon terminates earlier (**Fig 4D**). In all these examples, the insertion of the Pack-TYPE element likely occurred after the phylogenetic separation of the related species, supporting the idea that all Pack-TYPE TEs can have exon shuffling activity associated with their mobilisation.

We next checked if the models of these genes were consistent with their expression in aggregated RNA-seq experiments for maize and rice, available from the NCBI database (**S11 Fig**). In three cases out of four, we found evidence of transcripts merging exons originating from the Pack-TYPE TE and exons belonging to the ancestral gene locus (**S11B, S11C and S11D Fig**). In one case, although Zm-CACTA-181 is expressed in maize, its transcripts do not appear to be merged with the upstream gene (**S11A Fig**). These results suggest that Pack-TYPE TEs can contribute to the generation of new genes, but a Pack-TYPE insertion in a gene or its proximity does not necessarily lead to the formation of hybrid transcripts.

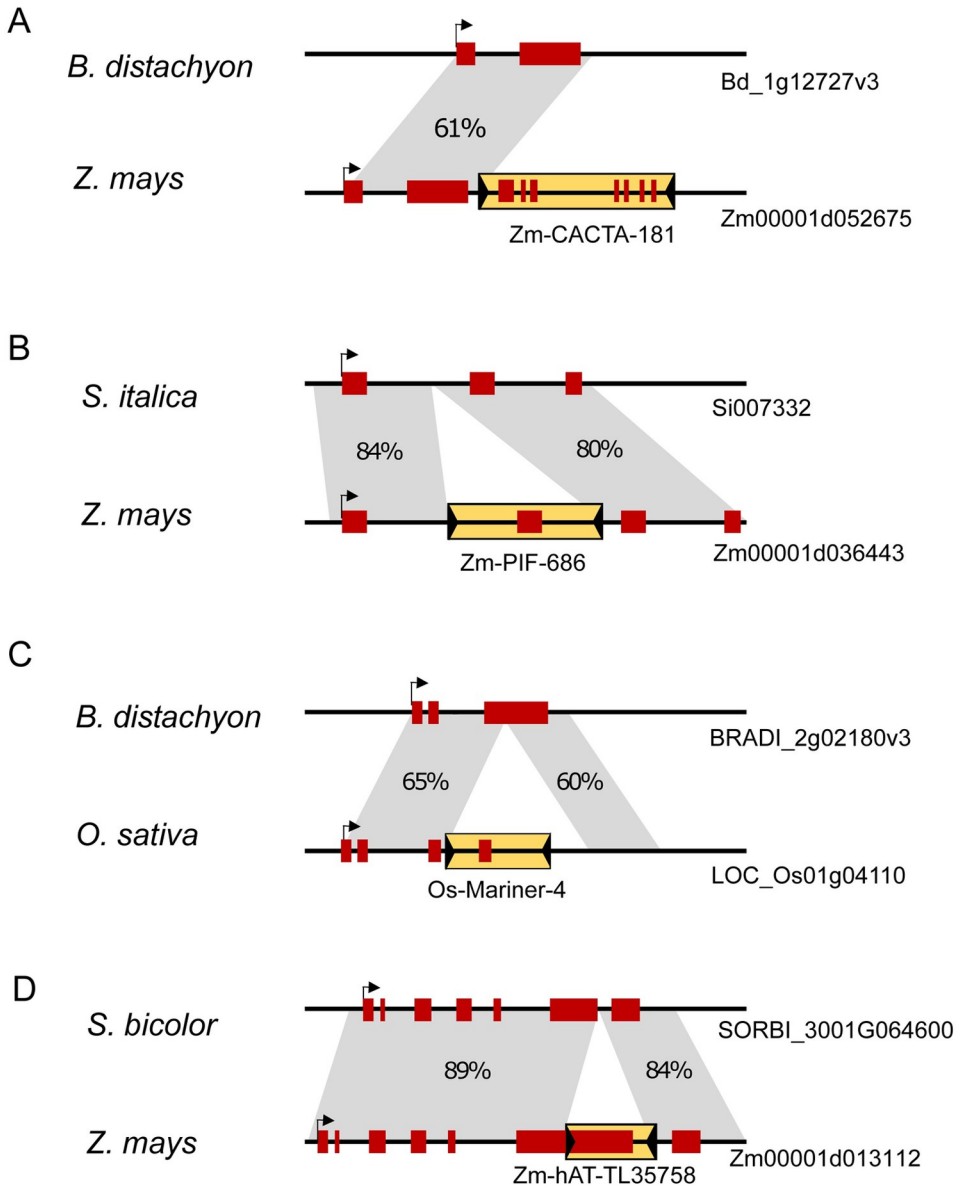

**Fig 4. Pack-TYPE TEs contribute to maize gene evolution. A** Scheme illustrating the maize Zm00001d052675 locus (with an insertion of a Pack-CACTA element) compared to the corresponding *B. distachyum* syntenic region. Sequence identity is displayed in the grey areas. The red blocks represent exons, and a black arrow indicates the start of transcription. The Pack-TYPE element is shown as a yellow box with black triangles representing TIRs. Exons model from a single transcript variant is displayed. **B** The Zm00001d036443 locus with the insertion of a Pack-PIF element, compared to a *S. italica* syntenic region. **C** The LOC_Os01g04110 locus with the insertion of a Pack-Mariner element, compared to a *B. distachyum* syntenic region. **D** The Zm00001d013112 locus with the insertion of a Pack-hAT element, compared to a *S. bicolor* syntenic region.

## Discussion

Several groups of TEs can mobilise and rearrange coding DNA in plant genomes [2], and there are several reported instances of gene transduplication events triggered by TEs [25]. However, many of these cases are associated with complex and atypical transposition events, believed to occur rarely in nature, which have been selected as a consequence of their positive effects on gene expression. Only two TE superfamilies are reportedly able to systematically

acquire coding DNA, including the Pack-MULEs described in rice [8,26] and *Helitron* TEs in maize [27]. Here, we provided evidence that not only *MULEs* but all main superfamilies of TIR-related TEs (i.e. *CACTA*, *Mariner*, *PIF* and *hAT*) can potentially generate Pack-TYPE elements in plants. The number of Pack-TYPE TEs we identified was variable in the maize and rice genomes and roughly correlated with the general abundance of TIR TE superfamilies annotated on each reference [28,29]. Generally, we observed a positive correlation between the total number of Pack-TEs detected in each group and their average length, which could be explained by the fact that longer elements are more likely to contain protein coding DNA.

Considering that most automatic annotation procedures for TIR TEs are homology-based [18,28], these are not optimised to detect TEs with more variable DNA and limited similarity to known TE reference libraries, which is a common feature of Pack-TYPE TEs. Indeed, we found that even in the *A. thaliana* TAIR10 annotation, only 2% of Pack-CACTA elements were correctly identified as elements belonging to the *CACTA* (*ATENSPM*) superfamily (**Fig 2**). Systematic studies of Pack-TYPE TEs have been reported only for Pack-MULE [9,11,26] as their relatively long TIR sequences facilitate their detection based on homology with existing *MULE* TEs [8,10]. Our approach uses a core TIR DNA sequence of 8 to 13 nucleotides as input; it relies on the conservation of TSDs and the presence of similar elements (clustering) as part of the detection process, allowing for the annotation of TE superfamilies with low TIR sequence conservation. With *packFinder*, we annotated a relatively small portion of known TIR TEs from reference genome annotations (**S6A Fig**). This is likely because: i) we used a limited set TIRs from well-known and abundant elements; ii) to increase confidence of TE detection and classification, TIR and TSD mismatches were kept to minimum; iii) we removed singleton elements. Due to the highly conservative nature of our approach, we have high confidence in *packFinder*-annotated TEs but we expect fewer elements to be detected compared to available TE annotations. However, *packFinder* also identified several elements missed in previous lists, suggesting that its integration with other tools could increase the completeness of TE annotations in plant genomes. In addition, contrary to most other TE annotation tools currently available, our detection pipeline is wholly embedded in an R/Bioconductor package [13] and can be easily performed using a single function, facilitating the detection and the study of Pack-TYPE transposons for any basic R user.

Besides Pack-TEs, we also annotated non-autonomous elements classified as non-Pack TEs; many of these belonged to the *Mariner* and *PIF* superfamilies in rice and the *PIF* and *hAT* superfamilies in maize (**Fig 2**). Considering that our annotation analysis bases the discrimination between Pack-TYPE and non-Pack TEs on a BLAST search, it is possible that Pack-TYPE elements with particularly diverged sequences are not annotated as such simply because a significant BLAST hit could not be found.

We observed similar distributions of Pack-TYPE, non-Pack and autonomous TEs for all superfamilies in both the maize and rice genomes, suggesting that they have similar insertion preferences. However, the distribution of historical TE insertions is not only a consequence of preferential insertions, but also depends by the rearrangement of DNA or the positive or negative selection of specific transposition events. In addition, Pack-TYPE TEs can potentially generate conflicts in the epigenetic regulation of the DNA portion with homology to genes, because small RNAs produced to silence TEs can target similar sequences found in actively expressed genes [30–32]. This phenomenon can lead to the fast pseudogenisation of less-constrained genes and rapid divergence of DNA sequences shared by functional genes and TEs [32], reducing the probability of significant blast hits. Therefore, BLAST approaches may only efficiently identify relatively recent acquisition events of protein coding DNA. Consistently, it has also been reported that the estimated age of Pack-MULEs in rice tend to be young, with elements remaining recognisable only for a few million years [9].

On the other hand, the genomic location of the original neighbouring insertions primarily determines the DNA captured by Pack-TYPE TEs and may not include coding DNA (**Fig 3A**). While previous studies on Pack-MULEs in rice suggest the preferential insertion of these elements into genes [9,10], it is also possible that at least a portion of the longer non-Pack TEs originated from capture events of intergenic chromosomal regions. In this case, these should be considered structurally similar to Pack-TYPE TEs. Nonetheless, considering most TEs classified as non-Pack are less than 500 bp in length (**Fig 2C**), a large proportion are likely constituted by just the two TIRs. Elements with such features are known as Miniature Inverted TEs (MITEs) and are abundant in plant genomes, primarily associated with the *PIF*, *Mariner*, and *hAT* superfamilies [33–35].

Both MITEs and Pack-TYPEs are common among TIR TEs and seem to share similar structures and transposition mechanisms, likely competing for the same transposases. Nonetheless, their abundance in the genome varies depending on the TE superfamily and plant species considered. Therefore, considering the role of Pack-TYPE TEs in gene transduplication, the balance in the relative mobilisation of MITEs and Pack-TYPE TEs could directly affect plant genome plasticity and the speed of gene evolution.

## Methods

### Reference genomes

The *Arabidopsis thaliana* TAIR reference genome (GCF_000001735.4), the *Oryza sativa* IRGSP-1.0 reference genome (GCF_001433935.1) and the *Zea mays* B73 reference genome (GCF_000005005.2) were downloaded from NCBI. In addition, general TE annotations for the *A*. *thaliana*, *Zea mays* and *Oryza sativa* genomes were obtained respectively from TAIR10 (https://www.arabidopsis.org/), the maize B73 annotation (https://mcstitzer.github.io/maize_TEs), and the rice IGRSP1 annotation project (https://rapdb.dna.affrc.go.jp).

### Automatic detection of Pack-TYPE transposons (*packFinder*)

To automatically detect Pack-TYPE TEs, we implemented an algorithm in three steps (**Fig 1A**). In the first step, we used two sets of TIRs. The first set was constituted of conserved TIR reference sequences at the terminal ends of well characterised autonomous TEs to survey the reference genome and find matches (**Tab C in S1 Table**). The second, extended set was obtained using TIR-Learner [18] to produce a list of the thirty most common TIRs in the rice and maize genomes, for each of the main TIR superfamilies investigated (**Tab G in S1 Table**). In order to get only the best-annotated TIR sequences, we ran only the first module of TIR-Lerner (Homology based annotation) on both rice and maize references (using default parameters for both genomes).

Base pair mismatches and indels are allowed by calculating Levenshtein distance to find optimal local matches [36]. We report the reference TIRs used and the number of allowed mismatches for each analysis in **Tab C in S1 Table**. The genomic fragments delimited by TIR pairs found close within the genome (in the interval of 300–15,000 bp, except for the extended set that used 300–5,000 bp) and in inverted orientation were selected as tentative TEs. This list was then filtered based on the presence of a TSD and its similarity at the 5' and 3' of the putative TE, again calculated using Levenshtein sequence distance (**Tab C in S1 Table**). Duplicated or overlapping TIR reference sequences can be defined due to the use of multiple TIR reference queries (**Tab C in S1 Table**); these were identified using the GenomicRanges [37] package and removed to prevent overestimation of TE abundances.

Subsequently, in the second step, we clustered (centroid based) tentative TEs using VSEARCH [38] to group TEs belonging to the same family. VSEARCH (default parameters)

clustered DNA sequences together where the edit distance was greater than 60%. To ensure quality alignments and stringent selection of putative TEs, we removed sequences with >10% wildcard nucleotides (N). To control for false positives and generate reliable lists of potential TEs, we removed singleton clusters assuming that real TEs are repeated in the genome. We also assumed that elements inside each cluster are evolutionarily linked, excluding the possibility of independent capture events of the same chromosomal sequence in different TEs (that should be rare). Therefore, we consider the cluster to be the closest approximation of a TE family in this work. Considering that the orientation of non-autonomous TEs cannot be defined easily, we arbitrarily considered the largest element of each cluster as forward orientated, and we determined the relative direction of the remaining TEs belonging to the same cluster by alignment.

The final list of putative TEs could include autonomous and non-autonomous TEs, and Pack-TYPE or not Pack-TYPE elements. Therefore, we applied a third step to automatically classify the putative TEs and identify potential Pack-TYPE TEs, according to the categories previously applied to Pack-MULEs [10]. Specifically, we used the CDS from the relevant reference genome (*Arabidopsis thaliana*, *Zea mays* or *Oryza sativa)* and the BLAST tool [39,40] to identify elements matching the sequence of transposase proteins. Any putative TE with a BLAST match (E-value < 1e-5, length > 250bp) to a transposase was automatically classified as an "autonomous TE" and was considered to be autonomous or derived from an autonomous element. We categorised the remaining elements as Pack-TYPE or non-Pack-TYPE based respectively on the presence or absence of BLAST hits (E-value < 1e-5, length > 50bp, except for the extended set that used length > 250bp) to a valid CDS entry. In this step BLAST was run on CDS databases with the options -max_target_seqs 500, -task blastn-short and -word_size 7 [7]. For each annotated TE of a specific superfamily and genome, we assigned a numerical ID in the order of discovery. To generate a unique element identifier, we concatenate the genome, the superfamily of the TIR sequence used for detection and the assigned ID (e.g. "At-CACTA-5" or "Os-Mariner-12"). We implemented the entire procedure in an R package called *packFinder* [13], which is publicly available as part of the Bioconductor project [41]. The results presented here were generated using *packFinder* v1.2.0. The TE density plots were generated using Circos [42].

## Benchmarking *packFinder* using annotated Pack-MULEs

We obtained the first 30bp of the terminal sequences of each *MULE* subfamily published by Ferguson *et al.* [10]; these sequences were used as input to *packFinder* to annotate elements in the rice genome. We then compared the output of *packFinder* to the 2776 TIR Pack-MULEs found previously with widths between 300–15,000bp [10]. For the purposes of testing the performance of the identification algorithm, only the first step of *packFinder* was applied (the BLAST and clustering steps were skipped).

To generate a receiver operating characteristic curve, we screened for Pack-MULEs with an allowable TSD mismatch of 2 and variable values of TIR mismatches. We then repeated this search using the forward TIRs as reverse TIRs and the reverse TIRs as forward TIRs. We subsequently calculated the true-positive rate (sensitivity) and false-positive rate (1 –specificity) as follows:

$$sensitivity = number\ of\ TIR-based\ Pack-MULEs\ identified/total\ TIR-based\ Pack-MULEs\ (2776)$$

$$1 - specificity = number\ of\ TEs\ found\ with\ reversed\ TIRs/number\ of\ TEs\ found\ with\ correct\ TIRs$$

### Analysis of neighbouring elements

For each cluster of TEs, we calculated the proportion of cluster elements within 100kb of another member of the cluster and compared this to the proportion of all non-member TEs within 100kb of a member of the cluster. We then tested the two proportions using a one-sided Chi-squared test for each TE superfamily. The directionality of annotated TEs was defined by their orientation relative to the largest element of their respective clusters.

### TIR clustering

We obtained the first 80 bp of all identified Pack-TYPE TE sequences, representing the forward TIRs, and used these to generate a distance matrix, using alignment-free kmer counting and a kmer size of 5 (default) [43]. We then applied quantile-based colour breaks to visualise the matrix as a heatmap and used complete-linkage hierarchical clustering to order the columns and rows of the distance matrix.

### Mappability

Mappability can estimate the repetitiveness of a TE in a genome [44]. Here, we computed mappability for both the rice and maize genomes with the GemMap tool [45], using a size of 20 and allowing a maximum of one mismatch. We converted the GemMap output to bigwig format in R with the package rtracklayer [46]. We then used deepTools [47] to plot the distribution of mappability averaged for each TE group, using a bin size of 20 nt. We obtained the general annotation of maize *CACTA*, *PIF* and *hAT* TEs using the B73 maize repeats annotation filtered for elements classified as "DTC", "DTH", and "DTA", respectively. The annotation of rice *Mariner* TEs was obtained by extracting "Stowaway" elements from the IRGSP1 repeats annotation [28].

### Expression analysis

We obtained the IDs for the genes that had significant BLAST matches to the transposons annotated in either the original or expanded set of Pack-TYPE TEs annotated in this work. Using these lists, we aimed to identify enriched functions and tissue localisations of the genes captured by these transposons in both rice and maize. Expression data for maize were obtained from ArrayExpress (E-MTAB-4342: https://www.ebi.ac.uk/arrayexpress/experiments/E-MTAB-4342/) [23], while for rice we obtained gene expression data from the MSU project (http://rice.uga.edu/expression.shtml) [22]. We filtered these datasets by genes captured by Pack-TYPE TEs before visualising the scaled expression data as heatmaps for each TE superfamily. Complete-linkage hierarchical clustering was used to order the columns and rows of the heatmaps.

To map maize genes to gene ontology (GO) terms, we used the maize-GAMER annotations for the B73 RefGen_V4 genome (https://data.nal.usda.gov/dataset/maize-gamer-go-annotations-methods-evaluation-and-review) [48] and the MSU project to map rice genes to GO terms [22]. The topGO R package [21] was used to carry out the enrichment analysis for each GO ontology (Biological Process, Molecular Function, Cellular Compartment). The enrichment analysis was performed for each TE superfamily for the same genes previously visualised in heatmaps. We used a Fisher test to determine the enrichment of GO terms with a minimum of five matches to captured genes; terms were treated as significant if they had P-values (adjusted by topGO's "weight01" algorithm) of less than 0.05.

### Synteny

We selected Pack-TEs overlapping gene coding regions using the R GenomicRanges package [37]. Then, syntenic genomic regions were identified using the EnsemblPlants Comparative

Genomics tools (https://plants.ensembl.org/index.html) in phylogenetically related species. The DNA sequence of the relevant loci were downloaded and aligned with Geneious (https://www.geneious.com) to generate alignment scores.

## Supporting information

**S1 Fig. Benchmarking *packFinder* against previously annotated Pack-MULEs.**
(PDF)

**S2 Fig. The whole-genome distribution of annotated TEs.**
(PDF)

**S3 Fig. Annotation of autonomous and non-autonomous TIR TEs.**
(PDF)

**S4 Fig. Repetitiveness of annotated Pack-TYPE-TEs.**
(PDF)

**S5 Fig. Examples of sequence structure for Pack-CACTA and Pack-hAT TEs.**
(PDF)

**S6 Fig. The whole-genome distribution of TEs annotated using TIRs derived from TIR Learner.**
(PDF)

**S7 Fig. The whole-genome distribution of TEs annotated using TIRs derived from TIR Learner.**
(PDF)

**S8 Fig. TIR relationships of TEs annotated using TIRs derived from TIR Learner.**
(PDF)

**S9 Fig. The properties of TEs annotated using TIRs derived from TIR Learner.**
(PDF)

**S10 Fig. Tissue-specific expression of genes captured by Pack-TYPE TEs.**
(PDF)

**S11 Fig. Expression of Pack-TYPE TEs in maize and rice.**
(PDF)

**S1 Table.  Tab A.** CACTA-like elements identified in *Arabidopsis thaliana*. **Tab B.** Pack-MULEs identified in rice by *packFinder*. **Tab C.** TIR sequences and parameters used to identify Pack-TYPE elements. **Tab D.** Elements identified with *packFinder* in *O. sativa* and *Z. mais* genomes using the TIRs in Tab C in S1 Table. **Tab E.** Number of elements identified by *packFinder* and their functional designations. **Tab F.** TEs used to produce the phylogenetic trees displayed in Fig 3. **Tab G.** The most abundant core TIR sequences (10 bp) in the maize and rice genomes (obtained with TIR-Learner), used to identify the expanded set of Pack-TYPE TEs in Tab H in S1 Table. **Tab H.** Elements identified with *packFinder* in the *O. sativa* and *Z. mais* genomes using TIR-Learner TIRs (Tab G in S1 Table). **Tab I.** Number of non-overlapping maize and rice loci captured by each Pack-TE identified by BLAST. **Tab J.** Enrichment analysis of the genes captured by Pack-TYPE TEs. **Tab K.** Number of Pack-TYPE TEs overlapping genes.
(XLSX)

## Acknowledgments

Part of the computations described in this paper were performed using the University of Birmingham's Compute and Storage for Life Sciences (CaStLeS) service.

## Author Contributions

**Conceptualization:** Marco Catoni.

**Data curation:** Jack S. Gisby, Marco Catoni.

**Funding acquisition:** Marco Catoni.

**Investigation:** Jack S. Gisby, Marco Catoni.

**Methodology:** Jack S. Gisby, Marco Catoni.

**Software:** Jack S. Gisby.

**Supervision:** Marco Catoni.

**Validation:** Jack S. Gisby.

**Writing – original draft:** Jack S. Gisby, Marco Catoni.

**Writing – review & editing:** Jack S. Gisby, Marco Catoni.

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
