## [Decision Letter · Decision Letter 0]

24 Sep 2021

Dear Dr Catoni,

Thank you very much for submitting your Research Article entitled 'The widespread nature of Pack-TYPE transposons reveals their importance for plant genome evolution' to PLOS Genetics.

The manuscript was fully evaluated at the editorial level and by independent peer reviewers.  I apologize for the lengthy review process in his case as we needed to accommodate some issues for the reviewers.  The reviewers appreciated the attention to an important problem, but raised some substantial concerns about the current manuscript. Based on the reviews, we will not be able to accept this version of the manuscript, but we would be willing to review a much-revised version. We cannot, of course, promise publication at that time.

The reviewers each had positive comments about the approach employed in this manuscript. There are some questions about the specific details and some concerns about some of the comparisons to prior annotations as in some cases these ignored relevant TE annotations in both species. Most importantly though was the consistent theme from the reviewers to extend the biological implications arising from the annotation of these elements. The reviewers made several suggestions for potential analyses that could add to the value of the manuscript. While it is not necessary to pursue all of the suggested questions it would be important to address some of these suggestions and add more information on the biology of these TEs and their impact on genome evolution.

If you decide to revise the manuscript for further consideration at PLOS Genetics, please aim to resubmit within the next 60 days, unless it will take extra time to address the concerns of the reviewers, in which case we would appreciate an expected resubmission date by email to plosgenetics@plos.org.

[LINK]

We are sorry that we cannot be more positive about your manuscript at this stage. Please do not hesitate to contact us if you have any concerns or questions.

Yours sincerely,

Nathan M. Springer

Associate Editor

PLOS Genetics

Claudia Köhler

Section Editor: Plant Genetics

PLOS Genetics

Reviewer's Responses to Questions

**Comments to the Authors:**

Reviewer #1: The manuscript by Gisby and Catoni describes the characterization of a new type of Transposable Elements (TE) : DNA transposons (class II) that are able to capture host DNA (including protein coding) sequences. The authors previously reported the existence of Pack-CACTA elements in Arabidopsis. Here, applying an automated pipeline they have developed to detect Pack-TYPE elements, they discovered a vast repertoire of Pack-TYPE elements in rice and maize reference genomes. This finding will have an impact on future standard TE annotation in plant genomes and the pipeline is likely to be incorporate in existing TE annotation pipelines. With the extent of large comparative genomic studies using long read sequencing data, the role of this new type of elements in structural variation is very likely to be commonly described in the near future.

The manuscript is well written and the results are clearly described. I have a main questions that the authors might answer easily:

- The authors mention that "all Pack-TYPEs tested contain less repeated DNA than all TEs annotated in rice and maize belonging to the corresponding superfamily (Figure S3)" on page 9: given the high TE content of the maize genome does this suggest that Pack-TYPE TEs arise in rather gene-rich regions? Apart from MITEs is there a type of class II that has a tendency to insert into gene-rich regions?

Out of curiosity I have two more questions:

- Do the authors have evidence that Pack-TYPE TEs display intraspecific polymorphisms ?

- Are the gene captured expressed ? (for instance from Fig 4?).

Minor comments:

I wonder if Mariner, hAT etc. should be italicized.

Page 11: "maize Zm-CACTA-181 TE is located in the 5-prime region of the gene Zm000021d052675": isn't it the 3 prime region?

Reviewer #2: In the manuscript “The Widespread nature of Pack-TYPE transposons reveals their importance for plant genome evolution”, Gisby and Catoni present a new method called PACKFinder to annotate PACK-type TEs using structural features of different superfamilies in plant genomes. When applied to Arabidopsis, their annotation method identifies several previously un-annotated non-Pack TEs, and Figure 1 shows that PACKFinder better captures the structural features of TIR elements than the established TE annotation. Then, the authors run PACKFinder on rice and maize, analyzing elements by superfamily, size, relationship, proximity, and ability to capture fragments.

When run on rice and maize, PACKFinder identified approximately 1,000 and 3,000 TIR elements in these genomes, respectively. These elements are a mix of PACK-type, autonomous, and non-autonomous elements of all major superfamilies of TIR DNA transposons. This number of elements is surprisingly small, especially given that the introduction cites the Jiang et al. 2004 paper, which identifies 3000 Pack-MULEs in rice. Then at the bottom of page 14, the authors point to URLs for maize and rice TE annotations that have an order of magnitude more DNA TE elements identified in the respective genomes. The authors do not compare PACKFinder TE annotations to the rice or maize annotations, so it is unclear why the number of identified elements is so different. Furthermore, the discussion section describes the benefit of PACKFinder as being a structure-based annotation method rather than homology-based, without comparing to the maize annotation that is already structure based.

In addition to typical considerations for TE annotations, Pack-type TEs require high-quality gene annotations to serve as a reference set to identify gene capture. In the methods, the authors state that Pack-TYPE was defined based on a BLAST hit > 50 bp to a “valid CDS entry”. This needs further clarification, especially since 50 bp is so short – quite a bit shorter than the average exon in either rice or maize. I am concerned here because the maize gene annotation used for this study is known for its poor quality, so it is possible that poor annotations could confound results here. In fact, I looked at the publicly available maize and rice genome browsers for the four genes shown in figure 4 and for two of them (in 4A and 4B), the alignment of RNA-seq reads does not support the figure. For example, in 4A, only the exons labeled Zm-CACTA-181 are expressed, so TE movement is not necessarily creating a new transcript. It would also be useful to know what kinds of genes are being captured by Pack-TYPE transposons, how many of them have known functions, and how many of them have syntenic orthologs with other grasses.

Another place where annotation issues could be interfering with the analysis is where the authors find that the median number of distinct loci within a Pack-TYPE transposon in maize is 3. This is very high and it seems improbable that the average element would have moved and picked up that many distinct pieces. I would like to see these captured fragments further analyzed to rule out the possibility that the fragments themselves are repetitive or part of a Mite that has inserted throughout the genome.

Overall, while it is valuable to have more bioinformatic tools specific to transposable elements that are easily accessible and easy to run on bioconductor, I would like to see more analysis performed on these annotations.

Reviewer #3: This is a review of the manuscript titled “The widespread nature of Pack-TYPE transposons reveals their importance for plant genome evolution” submitted by Gisby and Catoni. The authors develop a tool that is able to identify non-autonomous DNA transposons of various families (Pack-type), and apply it in rice and maize to assess the abundance, distribution and impact of these elements in plant genomes. Being a TE person myself involved in identifying and annotating TEs, this is certainly an interesting and helpful addition. There are very few class- or type-specific algorithms out there (HelitronScanner for Helitrons and MASiVE for Sireviruses as far as I know), as the vast majority annotate horizontally all DNA transposons or all LTR retros. Furthermore, developing a tool to identify TEs that do not have coding capacity is even more rare – and, importantly, even more difficult. I have my main concerns about this last point, i.e. how can such a tool be efficient with few false positives and at the same time able to identify most of the Pack-TYPE elements in a genome, which I discuss below.

First, however, I am confused with whether this paper is a technical paper or a research paper. Importantly, the pipeline has not been published and peer reviewed before (true?), which may give new ideas and bring up weaknesses that can substantially improve the identification process (this is what I mostly do here!). To my eyes, this is a technical paper, even though it is not set like that, with a little bit of more analysis offered: for example, the 3-4 genes with synteny, or the example of the internal structure of Packs (3C, 3E). It is not a research paper, as this would require a broader analysis to examine the extent of Pack’s in gene evolution, to quantify how many of these cases there are (this is very important, for now we are just looking at some few examples), to analyse if and how the captured or overlapping genes are affected (e.g. expression analysis, to see if they are hypothetical genes or have been annotated with some specific function). For example, I think the title is a bit too much on the biological side for what the paper gives to the reader. I am curious to see what the other reviewers and the editor thinks about that. I believe that some of the above suggestions are not hard to do, but it is not my call.

I think this is a technical paper and it is better if it is developed further in that direction. In this case, some of the Methods can be discussed and analyzed and tested further in Results. A schematic of the pipeline would help. Figure S2 should become a main. A Table with how many elements of each type were found should be made. My main concern however is about how well the pipeline works. We have no clear idea about false positives and how many Pack-type we are not finding. I think, based on my experience in this, that it is much safer to use only exons during BLAST; otherwise, one may detect hits with an intron, but if this intron is a TE, then noise and false positives are introduced. 58nt (or 50nt?) is too small. This is the result of the 1e-5 e-value, which is too soft. The distance of up to 15kb between the seed sequences is also too long. My approach would have been stricter. The TIR sequences in Table 1 are sometimes short, and even more so the TSD for Mariners. This is maybe evident in Fig. S2 for Mariners – how do the authors also explain the high k-mer distance in PIFs and CACTA (4th and 5th groups)? The papers cited in Table 1 for the TIR sequence are old. Are the any newer ones that may better describe these sequences and how the very between species?

All the above are not to say that this pipeline is not good; It can be a valuable addition to the TE toolbox. I am just trying to suggest that more discussion and thinking is needed on the design of the pipeline. Can the authors find a set of well annotated Pack-type TEs in a species and use it as a test to see how well it performs? This will be crucial if such a set exists. I am afraid that, as it is, is largely unknown how well it performs.

Some specific comments on the text:

--- 2nd paragraph of Intro ‘rearrange host genes’. Sometimes only gene fragments are captured and the original gene remains intact (if capture happened at the RNA level or if it repaired after capture). Sometimes now. I find this confusing, can the authors improve the description here? (same for ‘transduplication’ in Discussion.

--- 2nd paragraph of Results. #40 is not a Pack-type cluster.

--- 2nd paragraph of Results. About non-Pack TEs. How do you know that this is not simply some random genomic sequence? The annotation is non-Pack is based only on the proximity of short sequences (in reverse orientation) and a TSD which can be 2nt in Mariners. This extends to the discussion somewhere else in Results that some non-Pack elements may actually be old Pack-type. This may easily not be the case.

--- 2nd paragraph of Results. It is too strong to say that the TAIR10 annotation is not reliable. In some cases the Pack pipeline may be wrong, i.e. the gene annotation is correct – we do not know the extent of this.

--- Why did the authors not analysed Pack-MULEs as well? This good be a good dataset to compare how the pipeline performs?

**Have all data underlying the figures and results presented in the manuscript been provided?**

Reviewer #1: Yes

Reviewer #2: Yes

Reviewer #3: Yes

PLOS authors have the option to publish the peer review history of their article (what does this mean?). If published, this will include your full peer review and any attached files.

Reviewer #1: No

Reviewer #2: No

Reviewer #3: No

---

## [Decision Letter · Decision Letter 1]

28 Jan 2022

Dear Dr Catoni,

Thank you very much for submitting your Research Article entitled 'The widespread nature of Pack-TYPE transposons reveals their importance for plant genome evolution' to PLOS Genetics.

The manuscript was fully evaluated at the editorial level and by independent peer reviewers. Two of the reviewers felt that the revisions had fully addressed their concerns.  The third reviewer noted that while many concerns had been addressed there was still potential to improve the work by including a more explicit comparison to recent structural annotations of maize TEs. I do agree that this would be a useful addition to either the results or discussion section of the manuscript. 

We therefore ask you to modify the manuscript according to the review recommendations. Your revisions should address the specific points made by the reviewer along with a short response that details what has been changed.  I do not anticipate the need to have this sent out for peer review given the minor nature of the requested revisions.

[LINK]

Yours sincerely,

Nathan M. Springer

Associate Editor

PLOS Genetics

Claudia Köhler

Section Editor: Plant Genetics

PLOS Genetics

Reviewer's Responses to Questions

**Comments to the Authors:**

Reviewer #1: Dear authors,

Nice work, benchmarking Packfinder on rice PackMULEs was a very good idea and the results are convincing. The expression section is also very interesting and opens perspectives for our understanding of gene evolution.

Reviewer #2: Thank you for addressing many of my comments in this revision. Many of my major points were addressed and I think that the manuscript was improved with their additional analyses.

Specifically, the authors did a nice comparison of the PACKFinder TEs in rice to the existing homology-based annotation. As the authors pointed out, a structure-based annotation typically identifies fewer TEs of higher quality than a sequence homology-based approach.

However, the authors did not address the differences between their annotation and the maize annotation which is completely different from the rice annotation. The B73v4 TE annotation found at (https://mcstitzer.github.io/maize_TEs/)) is a structure-based TE annotation and still finds many more TEs than PACKFinder. Specifically, the B73v4 TE annotation finds 5,000 hAT, 2,700 CACTA, 63,000 PIF, 900 Mutator, and 66,000 Mariner transposons – far more than PACKFinder. What explaination is there for the widescale differences between PACKfinder and the existing maize annotation? For both maize and rice, what proportion of existing annotations overlap with PACKFinder TEs (like Fig S3 but from the opposite point of view)?

Since this is a methods-heavy paper, it would be helpful to include a paragraph breaking down similarities and differences to past annotations in the discussion to explain how this method improves upon prior work in the field.

Reviewer #3: I am now happy with what the authors did trying to explore how PackFinder performs.

**Have all data underlying the figures and results presented in the manuscript been provided?**

Reviewer #1: Yes

Reviewer #2: Yes

Reviewer #3: None

PLOS authors have the option to publish the peer review history of their article (what does this mean?). If published, this will include your full peer review and any attached files.

Reviewer #1: No

Reviewer #2: No

Reviewer #3: No

---

## [Editor Report · Decision Letter 2]

6 Feb 2022

Dear Dr Catoni,

We are pleased to inform you that your manuscript entitled "The widespread nature of Pack-TYPE transposons reveals their importance for plant genome evolution" has been editorially accepted for publication in PLOS Genetics. Congratulations!

Yours sincerely,

Nathan M. Springer

Associate Editor

PLOS Genetics

Claudia Köhler

Section Editor: Plant Genetics

PLOS Genetics

Comments from the reviewers (if applicable):

**Data Deposition**

http://datadryad.org/submit?journalID=pgenetics&manu=PGENETICS-D-21-00979R2

**Press Queries**

---

## [Editor Report · Acceptance letter]

21 Feb 2022

PGENETICS-D-21-00979R2 

The widespread nature of Pack-TYPE transposons reveals their importance for plant genome evolution 

Dear Dr Catoni, 

We are pleased to inform you that your manuscript entitled "The widespread nature of Pack-TYPE transposons reveals their importance for plant genome evolution" has been formally accepted for publication in PLOS Genetics! Your manuscript is now with our production department and you will be notified of the publication date in due course.

With kind regards,

Agnes Pap

PLOS Genetics

On behalf of:
